# High-Dose-Rate Brachytherapy for Treatment of Facial Skin Cancers: Local Control, Toxicity, and Quality of Life in 67 Patients

**DOI:** 10.3390/cancers16152742

**Published:** 2024-08-01

**Authors:** Jeanne Monge-Cadet, Benjamin Vairel, Mathilde Morisseau, Elizabeth Moyal, Anne Ducassou, Ciprian Chira, Cécile Pagès, Vincent Sibaud, Thomas Brun, Anouchka Modesto

**Affiliations:** 1Radiation Oncology Department, Oncopole Claudius Regaud and Institut Universitaire du Cancer de Toulouse, 1 Avenue Irène Joliot-Curie, 31059 Toulouse, Cedex 9, France; 2Surgery Department, Institut Universitaire du Cancer, 1 Avenue Irène Joliot-Curie, 31059 Toulouse, Cedex 9, France; 3Biostatistics and Health Data Science Unit, Oncopole Claudius Regaud and Institut Universitaire du Cancer de Toulouse, 1 Avenue Irène Joliot-Curie, 31059 Toulouse, Cedex 9, France; 4Dermato-Oncology Department, Oncopole Claudius Regaud and Institut Universitaire du Cancer, 1 Avenue Irène Joliot-Curie, 31059 Toulouse, Cedex 9, France; 5Physics Department, Institut Universitaire du Cancer, 1 Avenue Irène Joliot-Curie, 31059 Toulouse, Cedex 9, France

**Keywords:** skin cancer, brachytherapy, periorificial, toxicity, quality of life

## Abstract

**Simple Summary:**

Cutaneous squamous cell carcinoma and basal cell carcinoma are the most common cancers worldwide and surgical excision is the first-line treatment. Brachytherapy, which delivers a high dose of radiation to tumor tissue while sparing healthy tissue, represents a curative and conservative alternative in certain situations, particularly in elderly patients. Since the withdrawal of iridium wires from the market, brachytherapy has mainly been performed with high-dose-rate iridium-192 (HDR). Few studies and recommendations exist on HDR brachytherapy. Our study aims to evaluate the efficacy of HDR brachytherapy in terms of local control, survival, toxicity, and quality of life in patients with facial periorificial cutaneous squamous cell carcinoma or basal cell carcinoma treated in our center between 2015 and 2021.

**Abstract:**

While treatment of localized cutaneous squamous cell carcinoma (SCC) and basal cell carcinoma (BCC) is based on surgery, brachytherapy, which delivers a high dose of radiation to tumor tissue while sparing healthy tissue, is an alternative. Since the withdrawal of iridium wires from the market, brachytherapy has mainly been performed with high-dose-rate iridium-192 (HDR). This study evaluated the efficacy of HDR brachytherapy in terms of local control, survival, toxicity, and quality of life in patients with facial periorificial cutaneous SCC or BCC treated in our center between 2015 and 2021. Sixty-seven patients were treated for SCC (*n* = 49) or BCC (*n* = 18), on the nose (*n* = 29), lip (*n* = 28), eyelid (*n* = 7), or ear (*n* = 3). The majority had Tis or T1 tumors (73.1%). After a median follow-up of 28 months, 8 patients had a local recurrence. The local control rate at 3 years was 87.05% (95% CI 74.6–93.7). All patients developed grade 1–2 acute radio-mucositis or radiodermatitis and one experienced reversible grade 3 acute radio-mucositis. Of the 27 patients who completed the quality-of-life questionnaire, 77.8% recommended the treatment. This study confirms that HDR brachytherapy for facial cutaneous carcinomas provides good local control, good tolerance, and satisfactory functional outcome.

## 1. Introduction

Cutaneous squamous cell carcinoma (SCC) and basal cell carcinoma (BCC) are the most common cancers worldwide and incidence increases with age [1]. BCC is a lowly aggressive cancer with little risk of distant metastases, whereas there is an excess mortality rate due to SCC [2,3,4]. Surgical excision is the first-line treatment for newly diagnosed localized SCC and BCC, according to international recommendations [5,6,7]. To be complete, it must include margins that differ according to histological type [8], but it can lead to functional impairment or extensive reconstruction, albeit unsuitable in some elderly patients.

There are two main ways to administer brachytherapy on the skin: plesiotherapy or interstitial brachytherapy. Plesiotherapy, i.e., superficial brachytherapy, involves applying the radiation source directly on the skin. Leipzig or Valencia applicators, flaps and molds can be used according to the tumor size and localization. Interstitial brachytherapy may be used when the tumor is too deep. In this technique, the vectors are implanted directly in contact with the target tissue [9]. For several decades, interstitial brachytherapy delivered with iridium 192 has been an alternative therapeutic option associated with high local control and good cosmetic results in localized limited periorificial cutaneous cancers of the lips, eyelid, and nose. Brachytherapy delivers a high dose of irradiation to tumor tissue while sparing healthy tissue thanks to a strong dose gradient, and it represents a curative and conservative alternative in certain situations, particularly in elderly patients [10]. Our institution previously reported that locoregional-free survival was 87.3% at 5 years for periorificial skin cancer and relapse-free survival was 80% at 8 years for lip cancer with low-dose-rate (LDR) brachytherapy [11,12].

Since the withdrawal of iridium wires from the market in 2014, interstitial brachytherapy has been performed with after-loading machines. This has had consequences for clinical practice and, since then, we have been performing high-dose-rate (HDR) interstitial brachytherapy at our center for localized skin carcinomas patients eligible to brachytherapy. HDR brachytherapy delivers a dose rate greater than 12 Gy/h, unlike LDR brachytherapy which delivers a dose rate less than 0.2 Gy/h. Pulsed-dose-rate (PDR) brachytherapy is another modality which consists in short exposures to doses higher than those used in LDR brachytherapy (generally between 0.4 and 2 Gy/h). The advantage of using brachytherapy instead of external radiotherapy is the ability to deliver a higher dose to the skin with a steeper dose falloff.

This study evaluated the efficacy of HDR brachytherapy in terms of local control, survival, acute and late toxicity, and quality of life in patients with facial periorificial cutaneous SCC or BCC.

## 2. Materials and Methods

All consecutive patients treated by iridium-192 HDR brachytherapy at our center for a biopsy-proven periorificial skin carcinoma between 2015 and 2021 were eligible for this study. Data on demographics and clinical pathology were retrospectively extracted from prospectively collected medical records.

The project complied with ethical standards and the Helsinki Declaration of Human Rights was observed. Ethical committee approval was obtained (registration number: 22 RD 06) and was registered in the public directory of the Health Data Hub (F20220419095127).

The therapeutic strategy for each patient was discussed by the institutional dermatology or head and neck tumor board. All invasive SCCs tumors underwent locoregional imaging.

The choice of brachytherapy depended on size of the lesion, distance from bone structure, functional outcomes, and patient comorbidities and preference. Treatment was associated with a sentinel lymph node excision or cervical dissection for SCC. Elderly patients were evaluated using the G8 questionnaire and an onco-geriatric assessment was performed, if indicated [13].

Flexible implant tubes were implanted in the operating room under general or local anesthetic by a radiation oncologist with more than fifteen years of experience specialized in head and neck radiation therapy. For all patients, we used a flexible implant tube 5F single leader (Elekta, Stockholm, Sweden) with kind protection tube, placed inside the target volume with 7–8 mm spacing, in accordance with the Paris system rules, to ensure adequate coverage of the lesion with an additional margin of 0.5 to 1 cm depending on histology and location.

Sentinel lymph node search or neck dissection was performed during the same procedure by a head and neck surgeon in the event of T1 lip carcinoma or T2 skin carcinoma.

The patient then underwent a CT scan to perform dosimetry and assess the clinical target volume adequate coverage. Since 2017, CTV has been routinely delineated. The catheters were digitally reconstructed. Treatment was planned with the Oncentra Brachytherapy treatment planning system (Elekta, Stockholm, Sweden), following the general rules of the Paris system. First, an automatic optimization on the basal points was performed from the catheter reconstruction for the dosimetry (prescription was based on the 85% isodose method) [14,15]. Then, a manual graphical optimization was used to adjust the dose under the direct supervision of a radiotherapist and physicist. For patients treated for lip carcinoma, leaded protective splints were used to protect the underlying mandibular arch from irradiation.

The prescribed dose was 40 Gy in 8 fractions delivered over 5 consecutive days by the microSelectron™ after-loader with an iridium 192 source (Elekta, Stockholm, Sweden). Two sessions per day were performed from Tuesday to Thursday, with a free interval of at least 6 h. The last session took place on Friday morning, before the equipment was removed and the patient discharged.

Follow-up assessments were conducted routinely at 2 months by the radiotherapist, followed by subsequent evaluations every 3 to 6 months during the first year. Thereafter, follow-up was yearly based on age and comorbidities. Acute and late toxicities were recorded in medical reports and by photographs and were graded retrospectively according to CTCAE v5.0 (Common Terminology Criteria for Adverse Events version 5.0). Quality of life was self-reported (UW-QOLv4). The UW-QOLv4 addresses several domains, such as pain, appearance, activity, leisure, swallowing, chewing, speech, shoulder function, taste, saliva, mood, anxiety, and overall quality of life. Responses are scored between 0, 25, 50, 75 and 100 points and are based on the seven previous days at the time of questionnaire completion. Lowe’s team suggested separating the questionnaire into two scales: the socio-emotional and the physical [16]. The Physical subscale score is the average of six domain scores: chewing, swallowing, speech, taste, saliva and appearance. The Social-Emotional subscale score is also the average of six scores: anxiety, mood, pain, activity, recreation, and shoulder function.

The main endpoint was local control. Secondary endpoints were acute and late toxicity, overall survival (OS), relapse-free survival (RFS), and quality of life. Data were summarized by median and range (min–max) for continuous variables, and by frequency and percentage for qualitative variables. Local control (LC) was defined as the time from the initiation of treatment until the local relapse, patients without local relapse being censored at the date of death or last follow-up. OS was defined as the time from the initiation of treatment until death, patients alive being censored at last follow-up. RFS was defined as the time from the initiation of treatment until the first relapse (local, lymph node or metastatic) or death, patients alive and relapse-free being censored at last follow-up. Survival rates were estimated using the Kaplan–Meier method with their 95% confidence intervals (95% CI). Univariate analyses were performed using the Cox proportional hazards model for continuous variables and the Log-rank test for categorical variables. All statistical tests were two-sided and a *p*-value < 0.05 was considered statistically significant. Statistical analyses were carried out using Stata software version 16 (StataCorp LLC, College Station, TX, USA).

## 3. Results

From January 2015 to December 2021, 67 patients were treated with HDR iridium-192 interstitial brachytherapy for SCC or BCC of the nose (*n* = 29), lip (*n* = 28), eyelid (*n* = 7), or ear (*n* = 3). The median age was 73 years (46–96), and most patients were male (53.7%). Most presented with Tis or T1 tumors (73.1%). The characteristics of the patients and their tumors are detailed in Table 1.

The median number of vectors implanted per patient was 4. The median dose received was 40 Gy, delivered for most patients over 8 sessions. The median biological equivalent dose (BED) was 60 Gy. The main data are shown in Table 2.

Among SCCs (*n* = 28), 57.1% underwent sentinel lymph node screening or neck dissection. The initially planned sentinel node excision could not be performed in 2 patients owing to technical issues. In 48 patients (71.6%), brachytherapy was the first-intent treatment, and in 10 patients (14.9%) it was administered to treat a recurrence after resection. Nine patients received brachytherapy after suboptimal resection (13.4%). In 3 patients, neck dissection revealed nodal infiltration, so additional treatment by radiotherapy (*n* = 1) or radio-chemotherapy (*n* = 1) was administered. The last patient underwent a bilateral lymph node dissection owing to a positive sentinel lymph node biopsy.

Regarding dosimetric values, the median volume of the prescribed dose V_ref_ was 7.75 cc (1.21–32.34) (*n* = 67). The CTV was delineated in 38 patients and the median volume of the CTV delineated V_CTV_ was 4.33 cc (0.57–18.05), the median D90% was 4.93 Gy (2.23–10.75) per fraction and the median D98% was 3.68 Gy (1.75–9.21) per fraction (*n* = 38). The median CTV volume receiving the prescribed dose V_CTVref_ was 3.83 cc (0.39–15.26).

For the 38 patients, Figure 1 shows dosimetric values according to D90% and D98% and, on Figure 2, CTV coverage (Figure 2A) and volume variation (Figure 2B) are represented by boxplots.

During hospitalization, 6 patients experienced intercurrent events: agitation (*n* = 2), increased anxiety–depressive disorder (*n* = 1), eyelid hematoma (*n* = 1), cardiac arrhythmia (*n* = 1), and immediate post-operative facial paresis (*n* = 1). These events did not affect the length of hospital stay or the procedure, except for one patient who failed to receive the initially prescribed dose. Owing to a confusional episode, he tore off his equipment before the end of treatment and only 35 Gy were delivered instead of the 40 Gy initially planned.

After a median follow-up of 28 months (95% CI 19.1–32.5) on the 67 patients, 8 patients (11.9%) developed a local recurrence, 3 patients a nodal recurrence (4.5%) and 3 a metastatic recurrence (4.5%).

At 3 years for all patients, Local control was 87.05% (95% CI 74.60–93.65) (Figure 3), estimated OS was 79.54% (95% CI 63.91–88.95) and RFS was 70.99% (95% CI 55.84–81.75). Moreover, T2 tumors seemed to have a poorer local control compared to patients with T1 tumors (3y-LC: 72.2% (95% CI 35.3–90.3) vs. 92.2% (95% CI 77.8–97.4), respectively, HR: 3.95 (95% CI 0.87–18.05), *p* = 0.056). Then, patients with T2 tumors had a significantly lower RFS compared to patients with T1 tumors (3y-RFS: 36.8% (95% CI 7.2–68.4) vs. 79.3% (95% CI 62.3–89.3), HR: 2.99 (95% CI 1.09–8.19), *p* = 0.026). This trend was similar in OS (3y-OS: 50.6% (95% CI 14.0–79.0) vs. 84.7% (95% CI 66.8–93.4), respectively, for T2 and T1 tumors, HR: 3.20 (95% CI 0.93–11.05), *p* = 0.052).

In SCC (*n* = 49), local control at 3 years was 85.4% (95% CI 70.1–93.2), OS at 3 years was 77.5% (95% CI 59.6–88.2) and RFS at 3 years was 67.9% (95% CI 50.6–80.3).

For patients with BCC (*n* = 18), 3-year rates were 91.7% (95% CI 53.9–98.8) for LC, 87.5% (95% CI 38.7–98.1) for OS and 80.2% (95% CI 40.3–94.8) for RFS. BCC patients seemed to be associated with better OS (HR: 0.33 (95% CI 0.04–2.60), *p* = 0.269) and better RFS (HR: 0.55 (95% CI 0.16–1.91), *p* = 0.341) compared to SCC patients, but the difference was not significant and no difference was found for LC (HR: 0.94 (95% CI 0.19–4.69), *p* = 0.944).

All patients experienced grade I or II acute radio-mucositis or radiodermatitis, and one patient experienced reversible grade III acute radio-mucositis (1.6%). No severe late toxicity (after 6 months) was observed (Figure 4). One patient with an initial nasal SCC developed asymptomatic septal perforation after 6 months of follow-up and one developed an asymptomatic skin fistula on the treated area of the nose tip.

Among 56 alive patients, 27 (48.2%) completed the quality-of-life questionnaire. Of the patients completing the questionnaire, 77.8% recommended the treatment (21/27). The median score on the UW–QOL socio-emotional and physical scales was 95.8 (40.8–100) and 95.8 (61.7–100), respectively.

## 4. Discussion

This study is one of the largest to evaluate local control and toxicity after HDR brachytherapy for periorificial facial skin carcinoma including a quality-of-life questionnaire. Three-year local control rate was 87.3% with high patient satisfaction, which is in line with previously reported outcomes. Although the proportion of T2 lesions was high (21%), results were satisfactory in terms of oncological and cosmetic outcomes. No significant difference in survival and local control could be highlighted between SCC and BCC patients, probably due to a lack of statistical power, since there were few patients with BCC.

Only one patient experienced an acute grade III toxic event: reversible radio-mucositis at 6 months. Moreover, only one patient presented chronic toxicity: nasal septum perforation with no impact on his daily life. These findings are consistent with those of previous studies and show that brachytherapy is well-tolerated. Several retrospective series on interstitial brachytherapy of cutaneous cancers of the face have been published (Table 3). Survival data are similar, with a high local control rate in all series.

The 2018 GEC–ESTRO recommendations for skin brachytherapy do not contain any clear guidelines regarding interstitial HDR brachytherapy dosage. A high dose per fraction is recommended twice a day with at least 6 h between fractions. According to the tumor size and localization, a dose between 2.5 and 4 Gy should be used [17]. Recently, the GEC–ESTRO published a review of recommendations for skin superficial brachytherapy with flaps and customized molds but there was no mention of interstitial brachytherapy [18].

Although cosmetic outcomes are important, we believe overall quality of life is of paramount importance. Brachytherapy appears to preserve quality of life, although this issue has received little attention until now [19]. Our patients reported a high level of satisfaction in this regard.

The SCRIBE study comparing brachytherapy with radiotherapy in SCC and BCC of the skin demonstrated that the former resulted in a better post-treatment cosmetic appearance [20].

In an international meta-analysis by Lee et al. in 2019, a comparison was made between conventional excision, Mohs surgery, external-beam radiotherapy and brachytherapy. The results indicated that local control was comparable at one year across all treatments. However, brachytherapy was found to enhance cosmesis and achieve the highest local control rate for T1–T2 skin cancers [21].

In 2021, in a small case series of 23 patients with a median age of 89.5 years and published by Ferini and al. [22], facial SCC and BCC were treated using 5 fractions of 7 Gy delivered twice a week by hypo-fractionated radiotherapy approach. This approach achieved a local control rate of 95.5% at 6 months post-treatment for all the patients. Acute grade 3 toxicity was reported in 13% of patients, suggesting manageable side effects. Notably, these patients did not require general anesthesia or prolonged hospitalization. In addition, the hypo-fractionated approach minimized the number of hospital visits, which is beneficial for elderly and frail patients. Brachytherapy also minimizes the need for patient travel, as the treatment is typically administered and completed within the same week [17,18].

Since the discontinuation of iridium wires, HDR brachytherapy has been the mainstay for treating skin cancer. It has several advantages: patients are not connected to a wire and isolated in a room for the duration of the treatment and the medical staff are not exposed to radiation. When comparing HDR versus LDR brachytherapy for the treatment of lip carcinoma, it appears that control rates and toxicities are similar, with a trend towards reduced late toxicity with HDR brachytherapy [11,12,23,24]. Although brachytherapy is well-tolerated, the hospitalization can lead to the decompensation of comorbidities in elderly patients. In our study, we observed several states of agitation, one episode of cardiac arrhythmia and one psychiatric decompensation. Nevertheless, brachytherapy remains a treatment of choice in the elderly [25]. In addition, the TNM staging system contains no mention of the therapeutic consequences of potentially disfiguring surgeries affecting the face. For this reason, the European Association of Dermato-Oncology (EADO) has proposed a staging system for BCCs that contains five categories according to the difficulty of treating them. It takes into account lesional characteristics such as size, number and presence of nearby critical areas [26].

This study has several strengths including a relatively large sample size, a diversity of sites treated, prolonged follow-up, and homogeneous management. In addition to its retrospective nature, our study presents some limitations: an elderly cohort with a median age of 74, numerous comorbidities and more than 20% of the cohort presenting lesions classified as T2. Interestingly, despite these negative clinical features, our study confirms that HDR brachytherapy for facial cutaneous carcinomas provides high rate of local control, and satisfactory functional outcome. Prospective studies are now required to establish the optimal protocol for treating facial periorificial cutaneous SCC and BCC.

**Table 3 cancers-16-02742-t003:** Main studies on interstitial brachytherapy for facial cutaneous carcinomas. (F: fraction; BT: brachytherapy).

	Dose	Number of Patients	Localization	Time of Follow-Up (Months)	Local Control	Toxicities
**Tagliaferri et al.** [27] 2022	Nose 44 Gy: 3 Gy per F (first and last F 4 Gy) Lip 45 Gy: 5 Gy per F Eyelid 49 Gy: 3.5 Gy per F	40	Nasal vestibule, lip, eyelid	24 (median)	94% at 3 years	G1/G2 in most of patients No G3/G4
**Mareco et al.** [28] 2015	42.75 Gy in 10 F (median)	17	Eyelid	40 (median)	94.1%	**Acute** Conjunctivitis (42%) Hematoma (46%) Radiodermitis G3 (12%) **Late** No G3
**Guinot et al.** [29] 2014	45 Gy in 9 F (median) for 67% of patients	102	Lip	45 (median)	86.3%	**Acute** Hemorrhaging mucositis G4 in about 50% of cases Mucositis G3 (50%) **Late** Mild fibrosis without functional or cosmetic impact G1–G2 (100%)
**Cisek et al.** [30] 2021	49 Gy in 14 F 45 Gy in 9 F	28	Eyelid	24 (mean)	Amount 97%	**Acute** Skin RTOG G3 (3%) Skin RTOG G2 (32%) Conjunctivitis G2 (3%) **Late** Eyelid deformity G2 (5%) Skin lesions RTOG G1 (80%)
**Renard et al.** [31] 2021	7 Gy, followed by 8 × 4 Gy for exclusive BT 9 × 4 Gy for post-operative BT	66 (71 BT courses)	Lip, temple/cheek, nose, eyelid, ear, canthus	15.5 (median)	Local recurrence 6% at 2 years	**Acute** Dermatitis G3 (5.9%) Mucositis G3 (4.4%) **Late** No G3/G4 Significant hypopigmentation (5.9%)
**Tuček et al.** [32] 2023	54 Gy in 18 F	32	Lip	45 (median)	96.9% at 5 years	**Acute** G1 dry desquamation (18.8%) G2 erythema (31.2%) G3 confluent moist desquamation (50%) **Late** G1 fibrosis (100%), G2 depigmentation (18%) G1 telangiectasia (16%)
**Oliveira et al.** [33] 2024	36 Gy to 40 Gy in 9 to 10 F	58	Lower Eyelid	44 (median)	95% in the adjuvant group and 100% in the radical group at 4 years	**Acute** G1/G2 dermatitis (44.8%) G3 dermatitis (1.7%) G1/G2 conjunctival hyperemia (31%) Eyelid edema (24.1%) **Late** G1 (56.4%) G2 keratitis (1.8%) G3 cataracts (14.5%)
**Monge-Cadet et al.** 2024	40 Gy in 8 F (median)	67	Nose, lip, eye, ear	28 (median)	87.3% at 3 years	G1/G2 in most of patients **Acute** G3 radio-mucositis (1.5%)

## 5. Conclusions

HDR brachytherapy for cutaneous carcinomas of the face provides very good local control, good tolerance, and satisfactory functional and cosmetic results as an alternative to extensive or functionally disfiguring surgery.

## Figures and Tables

**Figure 1 cancers-16-02742-f001:**
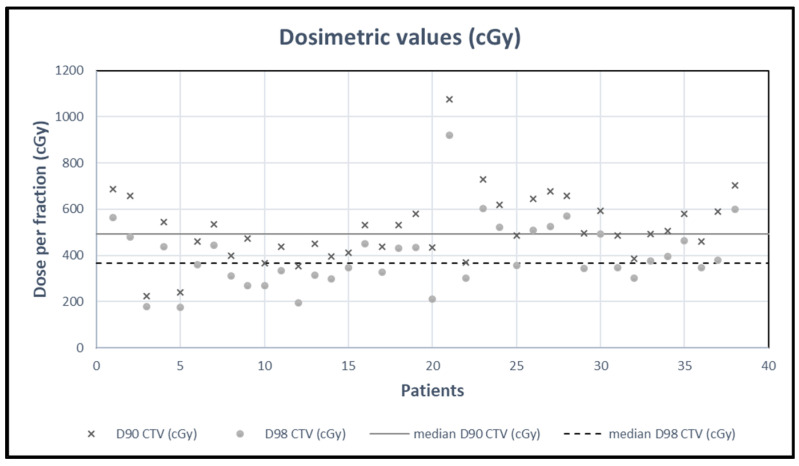
Dosimetric values (*n* = 38) for D90% and D98%.

**Figure 2 cancers-16-02742-f002:**
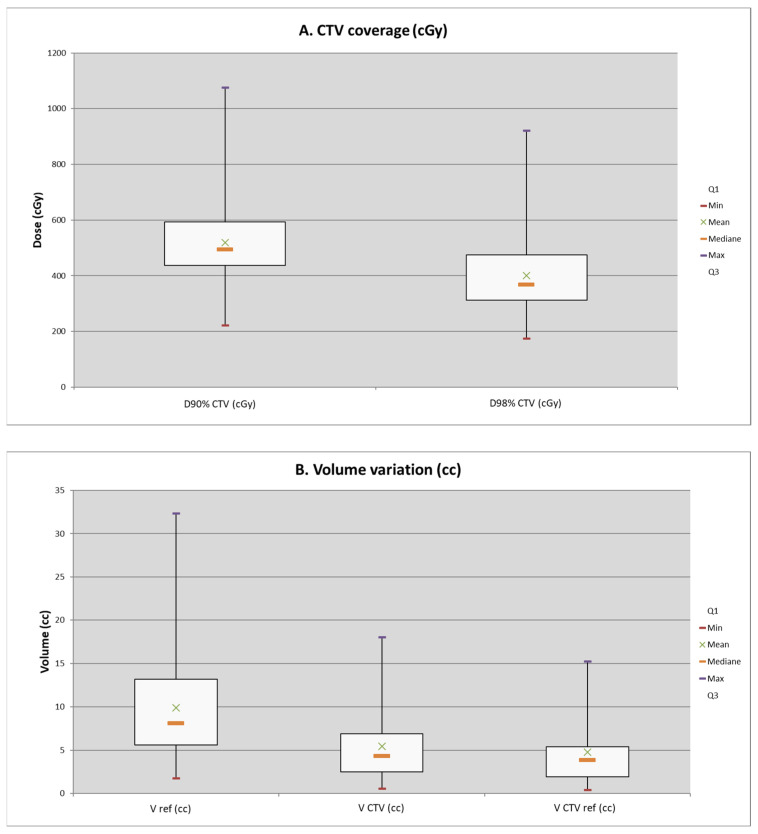
Boxplots of CTV coverage and volume variation (*n* = 38).

**Figure 3 cancers-16-02742-f003:**
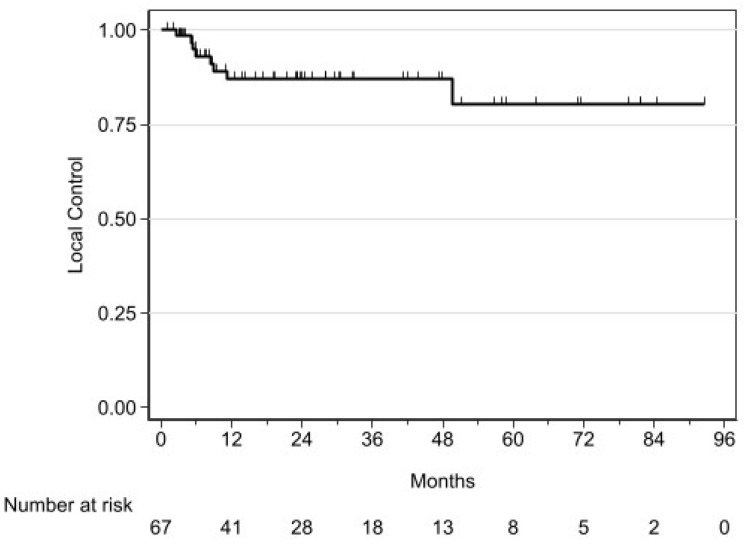
Local control (*n* = 67).

**Figure 4 cancers-16-02742-f004:**
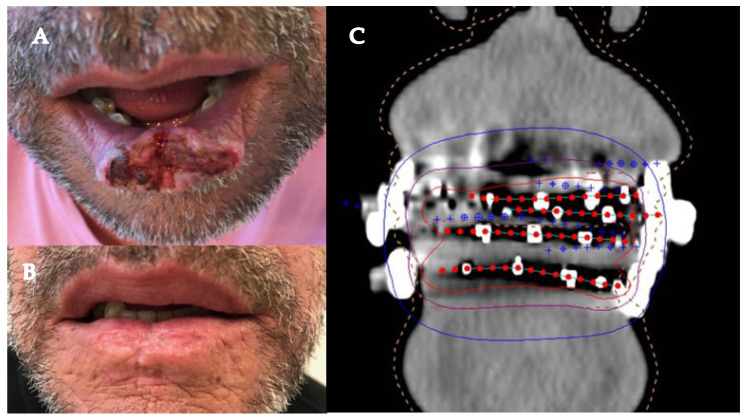
Patient 47 years old with a T2 SCC of the inferior lip. Photo before treatment with HDR brachytherapy (**A**) and 15 months later (**B**). Screenshot of dosimetry (**C**), blue crosses represent the reconstructed points, red points show active dwells, isodose of interest 250, 500 and 1000 cGy are defined in blue, purple and red, respectively.

**Table 1 cancers-16-02742-t001:** Characteristics of patients (*n* = 67). Performance Status refers to ECOG Performance Status Scale, which is scored from 0 (fully active) to 5 (dead) to evaluate patient’s daily living abilities. TNM refers to Classification of Malignant Tumors with T for tumor size, N for regional lymph nodes involved and M for distant metastasis involved.

Characteristics		N/Median	Min–Max/%
**Age (years)**		73	46–96
**Sex**	Women	31	46.3
	Men	36	53.7
**Performance Status**	0–1	47	78.3
	2–3	13	21.7
	Missing	7	
**Tobacco**	Yes	38	56.7
	No	29	43.3
**High blood pressure**	Yes	31	46.3
	No	36	53.7
**Diabetes**	Yes	8	11.9
	No	59	88.1
**History of cancer (all types)**	Yes	31	46.3
	No	36	53.7
**History of skin cancer**	Yes	13	19.4
	No	54	80.6
**Tumor location**	**Nose**	29	43.3
	Impairment of nasal septum	6	30.0
	**Lip**	28	41.8
	Commissure involvement	2	7.7
	**Eyelid**	7	10.4
	**Ear**	3	4.5
**Histology**	SCC	49	73.1
	BCC	18	26.9
**Aggressivity**	In Situ	3	4.5
	Invasive	64	95.5
**T**	1	49	73.1
	2	14	20.9
	X	4	6.0
**N**	0	42	80.8
	1	3	5.8
	X	7	13.5
	NA	15	
**M**	0	67	100

**Table 2 cancers-16-02742-t002:** Brachytherapy treatment (*n* = 67).

	Median Value/N	Min–Max/%
**Number of applicators**	4	1–9
**Dose (Gy)**	40	35–48
**BED (Gy)**	60	52.5–76.8
**Number of fractions**	8	7–9
**Anesthesia**		
**Local**	16	23.9
**General**	51	76.1
**Number of days in hospital**	5	5–5

## Data Availability

Data are unavailable due to patients confidentiality.

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
