# Peer review of "High-Dose-Rate Brachytherapy for Treatment of Facial Skin Cancers: Local Control, Toxicity, and Quality of Life in 67 Patients"

_cancers, 2024, doi:10.3390/cancers16152742_

Round 1
Reviewer 1 Report
Comments and Suggestions for Authors
In this study, Ir-102 HDR brachytherapy for facial skin carcinomas was evaluated for overall survival and relapse free survival. In general, the manuscript is well written and the results are encouraging for the use of brachytherapy as a treatment modality. This is an important paper since many centers offer brachytherapy for prostate, gynecological, and breast sites, but fewer offer options for treating skin carcinomas with HDR brachytherapy.
There were several editorial comments that need to be addressed prior to publication.
Line 99: What are vectors? I am not familiar with this object as it pertains to skin brachytherapy. Please provide a brief description of what they are.
Line 100: What is a senior radiotherapist? In the USA, a radiotherapist is usually referred to a radiotherapy technologist with two to four years of vocational training as opposed to a radiation oncologist who is a medical doctor. Please clarify this. I think the term radiotherapist should be changed to radiation oncologist to eliminate any confusion.
Table 1. For clarity, state what T, N, M are. I assume it means tumor size/stage, nodal involvement, metastasis. Unsure what Performance Status 0-1, 2-3, missing means.
Graphs: The label units for the ordinate and abscissa on many of the graphs
Lines 192-193: Are the numbers in brackets [], reporting percentages to be consistent with the results in the rest of the paragraph? This is unclear.
Reviewer 2 Report
Comments and Suggestions for Authors
Line 77 - With "smaller gradient dose", do you mean "steeper dose falloff"?
Lines 87-88 - " The methods were performed in accordance with relevant guidelines and regulations.", you should cite proper references supporting this sentence. In this guise, it is too vague.
Line 109 - "Prescription was based on the 85% isodose method". Please, clarify this method.
Line 130 - "... seven previous days ..." What does this mean? Why did you interview the patients about seven days before the questionnaire administration and not about the current day?
Line 140 - you also included the event "death" in RFS. This means that a patient who has died without relapse, even for cancer-unrelated cause, will be counted as relapsed rather than censored. Why this choice?
Lines from 164 to 167: You listed 3 patients. The patient who was not administered with RT or RCT was the one submitted to node dissection, wasn't he? Clarify this issue.
Lines from 168 to 171: Why did you refer to the "mean" values rather than the "median" ones?
In Figure 1, the caption should be more exhaustive. For example, I think that in the X axis each of the 38 patients is represented while in the Y axis, there are the doses per fraction expressed in cGy, correct?
In the caption of Figure 2, you should clearly describe what is represented.
In lines 193-195, the survival outcomes (LC and OS) for the SCC subgroup are slightly inferior to those of the overall cohort, thus suggesting better values for the BCC subgroup, as reasonably expected. Was this difference statically significant? Please, address this issue in the manuscript.
Lines from 211 to 216 (from "Lowe's..." to "... shoulder function") should be moved in the methods section.
Line 225: carcinologic??
In Table 4 in the first box citing the work by Tagliaferri replace "by F" with "per F".
General comments:
In the introduction you could cite PMID: 35454779 together with the references 2 and 3.
I would appreciate if you'll discuss the differences with other radiotherapy approaches like hypofractionated electron beam radiotherapy, which showed promising results in other elderly cohorts similar to yours, without anesthesiological risks and need for hospital stay. Please, refers to PMID: 33948304.
Comments on the Quality of English Language
Some minor editing needed.
Round 2
Reviewer 2 Report
Comments and Suggestions for Authors
Thank you for having addressed all my suggestions. Only two more issues to fix:
1) Specify that N in the TNM staging refers to REGIONAL lymph nodes
2) Replace "carcinological" with "oncological"
Author Response
Thank you for your time and your constructive comments.1) Specify that N in the TNM staging refers to REGIONAL lymph nodes
This has been modified in the latest version of the manuscript.
2) Replace "carcinological" with "oncological"
This has been modified in the latest version of the manuscript.